# A Study on the Impact of Institutional Pressure on Carbon Information Disclosure: The Mediating Effect of Enterprise Peer Influence

**DOI:** 10.3390/ijerph19074174

**Published:** 2022-03-31

**Authors:** Yongjun Tang, Jun Zhu, Wenchao Ma, Mengxue Zhao

**Affiliations:** 1Business School, Hohai University, Nanjing 211100, China; yjtang@hhu.edu.cn (Y.T.); 201313040007@hhu.edu.cn (J.Z.); 18763807230@163.com (M.Z.); 2School of Accounting, Zhejiang Gongshang University, Hangzhou 310018, China

**Keywords:** institutional pressure, carbon information disclosure, environmental regulation, Confucian culture, enterprise peer influence

## Abstract

Enterprises should bear the main responsibility for greenhouse gas emissions. Disclosing carbon emission information is one of the important ways for enterprises to deal with climate change. Taking China’s A-share listed companies from 2014 to 2018 as the research sample, we study the impact of external explicit institutional pressure and implicit institutional pressure on corporate carbon information disclosure and analyze the mediating effect of enterprise peer influence in carbon disclosure. The empirical results show that external institutional pressure, namely environmental regulation and Confucian culture, has a significant positive impact on enterprise carbon information disclosure. Enterprise peer influence has a certain mediating effect between external institutional pressure and carbon information disclosure. The government should formulate and improve the carbon information disclosure institution and strengthen external supervision through the joint participation of all sectors of society.

## 1. Introduction

At this stage, China’s social and economic development is showing a medium and high-speed growth trend, but one thing that cannot be ignored is that carbon emissions lead to increasingly serious air pollution and climate warming, which poses a great threat to China’s social and ecological environment security. This has naturally attracted the great attention of China's government and all sectors of society. In the 2015 Paris climate change agreement, the world reached an important consensus on curbing greenhouse gas emissions. Carbon emissions have gradually become a hot topic of widespread concern in the field of resources and the environment [1].

It is imperative to strengthen the management of resources and environmental protection. More and more institutional investors, media, and people from all walks of life have begun to pay attention to the environmental cost behind the development of enterprises. In the process of daily social operation, enterprises play a dual role. They are not only the main creator of China’s social and economic wealth but also the main body responsible for carbon emissions [2]; therefore, on the major issue of carbon emission reduction, enterprises should assume more social environmental protection responsibilities and improve the quality of carbon information disclosure [3]. Before 2022, most carbon information disclosure of Chinese enterprises belonged to the category of voluntary disclosure. On 8 February 2022, the administrative measures for the legal disclosure of enterprise environmental information were implemented. Key pollutant discharge units and other enterprises with great environmental impact and public attention shall disclose carbon emission information in accordance with the specified contents and time limit. The government’s supervision of corporate carbon information disclosure is more in-process and ex-post supervision. Enterprises that do not disclose environmental information or disclose untrue and inaccurate environmental information will be fined. By checking the enterprise carbon information disclosure report, the government reduces the “greenwashing” behavior of enterprises to a certain extent and improves the quality of enterprise carbon information disclosure. As far as the motivation of carbon information disclosure is concerned, enterprises need to invest a large amount of financial, material, and human resources to achieve energy conservation and emission reduction, which significantly increases the operating burden of enterprises and discourages the enthusiasm of enterprises to carry out energy conservation and emission reduction activities; therefore, the core motivation for Chinese enterprises to disclose carbon information is still to meet the requirements of government legitimacy under external institutional pressure.

External institutional pressure can be divided into explicit institution pressure and implicit institution pressure [4]. An explicit institution refers to the officially promulgated laws and regulations and various policies and standards formulated by the government, which can form a direct pressure on enterprises [5]. China proposes to accelerate the construction of a green and low carbon circular development economic institution, and achieve a carbon peak in 2030 and carbon neutralization in 2060. In recent years, China has made important amendments to the environmental protection law, formulated and passed the environmental protection tax law, and has issued a series of rules and regulations aimed at standardizing the carbon emission behavior of enterprises and promoting the green transformation and upgrading of enterprises. Implicit institution refers to the informal institution taking traditional culture as the core. Traditional culture has a subtle impact on the moral concept and responsibility consciousness of enterprises through common values [6]. Chinese traditional culture, represented by Confucian culture, advocates the unity of heaven and man, conforms to nature, and advocates the realization of harmonious coexistence between man and nature. The excellent character of protecting the natural environment and emphasizing social responsibility in the traditional culture will affect the green development plan of enterprise managers, guide them to actively carry out carbon emission reduction activities, and disclose carbon information truly and regularly; therefore, traditional culture can virtually regulate the carbon information disclosure behavior of enterprises [7].

Facing the increasingly strict external institutional pressure, in order to better survive in the fierce environmental competition and maintain their social image, enterprises will pay attention to the behavior of competitors in the same industry and instinctively engage in behaviors consistent with leading enterprises with large market share and great influence in the industry. Enterprises with stronger peer influence are more willing to disclose more detailed and useful carbon information to the outside world so as to meet the urgent needs of different subjects from all walks of life for carbon information, such as the government and regulatory authorities, investors and creditors, accounting firms and lawyer firms and other intermediaries, consumers, consumer protection organizations, and the public [8,9]. In the process of carbon accounting information disclosure, enterprises will be affected not only by their own characteristics but also by peers [10]. We define the reaction of an enterprise’s own behavior driven by paying attention to the changes of some behaviors of other enterprises in the industry in the process of industry development as peer influence in carbon information disclosure. In the process of carbon information disclosure, enterprises will choose to follow the behavior of peer enterprises in order to avoid the risk of uncertain results caused by resource constraints [11,12].

So far, most of the research on institutional pressure and carbon information disclosure in domestic and foreign academic circles still focuses on the impact of various formal, explicit institutions, and pays less attention to the impact of potential informal institutions [13,14]. Specifically, Shen and Feng (2012) believe that the government’s environmental supervision policy system can significantly improve the environmental information disclosure level of listed companies [15]. Through the analysis of institutional theory, Herold et al. (2018) discuss the generation and institutionalization of carbon disclosure and its impact on carbon reporting [16]. From the perspective of the government, Eka (2016) believes that the performance evaluation plan in environmental management is a tool to encourage and establish industrial enterprises’ compliance and awareness of environmental management regulations [17]. The carbon disclosure provided by enterprises includes not only economic information but also non-economic information. Grauel and Gotthardt (2016) find that environmental regulations and legal sources are very relevant explanatory elements to corporate carbon information disclosure [18]. Kalu et al. (2016) believe that society and the financial market are the key elements determining carbon information disclosure [14]. The improvement of public education and awareness will increase the social requirements for enterprise carbon information disclosure and the degree of enterprise carbon information disclosure. Tower et al. (2015) study the social and environmental information disclosure (SED) of Indian textile enterprises from 2010 to 2012 and find that the pressure from informal institutions has greatly promoted the social and environmental information disclosure of enterprises [19]. He et al. (2021) take Chinese-listed companies in the CSI 300 index from 2012 to 2018 as a sample and find that Confucianism had a positive impact on the quality of carbon information disclosure [20].

Supported by legitimacy theory, institutional theory, signaling theory, and sustainable development theory, we select the social responsibility report of China’s A-share listed companies as the research sample. From the perspective of different institutional pressures faced by enterprises, we systematically study the impact of explicit institutional pressure and implicit institutional pressure on enterprise carbon information disclosure under the mediating effect of peer influence. The research innovation is mainly reflected in the following three aspects: (1) Construct the evaluation system of carbon information disclosure level of Chinese A-share listed companies. Combined with China’s government regulatory requirements and the development direction of enterprises under the “double carbon” goal, and referring to the GRI standard, this paper establishes the carbon information disclosure evaluation system of Chinese listed companies. (2) Distinguish the types of external institutional pressure. We divide the external institutional pressure into explicit institutional pressure and implicit institutional pressure, which are measured by government environmental regulation and Confucian culture, respectively. This enriches the theoretical research on institutional pressure. (3) Innovation of research perspective. We introduce the mediating variable of enterprise peer influence. We fully integrate the internal and external factors of enterprises, and put institutional pressure, enterprise peer influence, and enterprise carbon information disclosure into the same research framework. Our research helps to interpret the impact mechanism of institutional pressure on enterprise carbon information disclosure from the perspectives of formal and informal institutions and provides policy suggestions for promoting enterprises to actively fulfill their environmental responsibility and improve the quality of carbon information disclosure.

## 2. Theoretical Analysis and Research Hypothesis

### 2.1. Institutional Pressure and Corporate Carbon Information Disclosure

With the successive release of relevant government policies and regulations on corporate environmental information disclosure, the public expects enterprises to fully and truly disclose carbon information, thus listed companies facing the external institutional pressure has also been increasing. From the perspective of legitimacy theory, it is possible that enterprises disclose carbon information to maintain the legitimacy of their behavior. The more detailed and sufficient the carbon information content in the responsibility report is, the more it will help the company obtain social recognition and thus maintain the legitimacy of the company’s business activities [21]. Under the action of institutional theory, China's central and local governments and departments will force the introduction of the relevant policy system from the perspective of the public interest and require the listed companies to disclose the relevant carbon information content in the production and operation process in real time, so as to avoid the possible information asymmetry between the stakeholders. At the same time, combined with the concept of environmental protection, the governments achieve the comprehensive treatment of carbon emission reduction [22].

The carbon information disclosure behavior of enterprises will be obviously affected by both external explicit institutional pressure and implicit institutional pressure [23,24]. On the one hand, from the perspective of explicit institutional pressure, when a local government issues clear guidelines and policies on environmental protection, it pays great attention to the environmental protection work of local enterprises. Then enterprises must consciously strengthen the supervision and management of their own carbon information disclosure and are urged to disclose legal, reasonable, comprehensive, and effective environmental information. In other words, the targeted and mandatory policies and systems issued by the government on enterprise carbon information disclosure have the most direct impact on the disclosure of such information [25]. In addition, when the government issues energy conservation and emission reduction policies, it will jointly formulate a series of supporting schemes with central and local financial institutions, such as tax incentives, financial subsidies, green finance, etc., to ensure the effective implementation of the policies. Enterprises often have to meet the minimum carbon information disclosure standards set by the government in order to meet the government’s support conditions. In recent years, the Chinese government has committed itself to the construction of ecological civilization, worked hard to protect the environment, performed well in energy conservation and emission reduction, and severely cracked down on environmental pollution by enterprises; therefore, the illegal cost of “greenwashing” is high. Once the “greenwashing” behavior is exposed, the enterprise will lose its foothold in the market. Under strong regulatory pressure, enterprises have to truly disclose their carbon emission information. In contrast, non-Chinese enterprises face less regulatory pressure. When there are loopholes in the regulatory system, the cost of “greenwashing” is often very low. In order to pursue economic profits, enterprises may implement “greenwashing” [26]. To sum up, under the supervision of the government and other third-party institutions, Chinese enterprises are less likely to have “greenwashing” and deception in the process of carbon information disclosure. In order to realize the legitimacy of sustainable operation, enterprises will choose to accept and fulfill the responsibility of carbon emission reduction, and the external institutional environment will drive enterprises to take the initiative to adopt the behavior of carbon information disclosure.

On the other hand, from the perspective of implicit institutional pressure, as the core of Chinese traditional culture, Confucian culture is an informal system with the most far-reaching impact on Chinese society. With “benevolence, righteousness, courtesy, wisdom, and faith” as the main content, it is an ideological and cultural system centered on cultivating and shaping personality [27,28]. Culture educates people and contains rich moral concepts. Education and governance complement each other. According to the viewpoint of new institutional economics, traditional culture can not only improve personal moral cultivation but also cooperate with explicit institutions to make up for the loopholes of explicit institutions in the process of restricting enterprise behavior and jointly affect the carbon information disclosure behavior of enterprises. This concept will have a subtle and positive impact on the environmental protection behavior of Chinese enterprises. First, Confucian culture advocates the idea of “the unity of heaven and man”, which holds that man and nature are harmonious and unified, and man and nature prosper together [29,30]; therefore, man should follow the laws of nature and live in harmony with nature [31]. This idea also has an impact on the environmental protection behavior of enterprises. Compared with the pure pursuit of the business objective of maximizing economic benefits, Chinese entrepreneurs tend to regard the protection of the natural environment as their own responsibility due to the concept of “harmony between man and nature” that has been passed down from generation to generation, and then actively disclose corporate carbon information under the pressure of this concept. Second, Confucian culture advocates the values of “righteousness and benefit” and “loyalty and faith”, and holds that reasonable people take righteousness as benefit and righteousness as the criterion for profit. Loyalty and faithfulness are two of the most important characteristics, which require people to be loyal to their family, country, and the world, and have the courage to bear social responsibility. Enterprises are economic organizations for profit; however, the production and operation activities of enterprises should not be at the cost of destroying the ecological environment. Enterprises should actively undertake social responsibility, achieve the unity of business development and environmental protection, and strive to realize green transformation. In addition, the Confucian thought of good faith helps to curb the false statements of enterprises and encourages enterprises to disclose the real carbon emissions of the company in a timely and objective manner [27]. In Chinese culture, honesty is not only the fundamental criterion of life but also the prerequisite for the survival of enterprises. In the spiritual system of modern business ethics in China, integrity occupies an important position, unobtrusively and imperceptibly affecting the fraudulent behavior of organizations and members and restricting their egoistic mentality. Under the strict supervision of the Chinese government and the edification of an integrity culture, the authenticity and objectivity of enterprises in the process of carbon information disclosure can be reasonably guaranteed [32]. Third, Confucian culture emphasizes “benevolence”. Its essence is love, and it can push itself to others and spread to the public. In order to achieve sustainable development, enterprises must fully consider the requirements of investors, the public and other stakeholders, timely and comprehensively disclose the impact of their carbon emission behavior, win the understanding and support of stakeholders, and establish the corporate image of green emission reduction [33]. Nowadays, the awareness of environmental protection in all sectors of society is constantly improving, which has brought great potential pressure to enterprises to a certain extent. In the face of this situation, enterprises will actively disclose carbon information in order to enhance their competitiveness and achieve sustainable development [19]. It can be seen that the motivation of enterprises to disclose carbon information is based on the dual role of explicit and implicit institutional pressure; therefore, we put forward the following assumptions:

**Hypothesis** **1**.
*The external explicit institutional pressure on enterprises has a significant positive effect on the improvement of carbon disclosure levels.*


**Hypothesis** **2**.
*The external implicit institutional pressure on enterprises has a significant positive effect on the improvement of carbon disclosure levels.*


### 2.2. Institutional Pressure and Enterprise Peer Influence

The peer influence of enterprises is affected by external institutional pressure. By analyzing the behavior of enterprises based on the theory of institutional pressure, it can be found that enterprises are currently facing pressure from all walks of life. Their behavior regarding carbon information disclosure is affected by different external institutional pressures, the sources of which are explicit institutional pressure and implicit institutional pressure [34]. From the perspective of explicit institutional pressure, government departments have issued some mandatory laws and regulations to supervise and standardize the carbon information disclosure of enterprises and severely punish those enterprises that do nothing in environmental protection. Increasingly strict environmental regulation has greatly stimulated the management of enterprises. In order to avoid environmental punishment, enterprises tend to improve their response-ability to external environmental changes. In order to avoid falling behind in business competition, enterprises will keep up with the carbon information disclosure level of enterprises in the same industry. According to the theory of new institutionalism, enterprises often imitate the structure and behavior of other organizations in the same environment to meet the requirements of legitimacy so as to alleviate external risks [10,26]. When the enterprise fails to keep up with the carbon information disclosure level of enterprises in the same industry, the enterprise is likely to become the object of public opinion attack, resulting in damage to the reputation of the enterprise and an adverse impact on the financing and operation of the enterprise. In addition, after keeping up with the carbon information disclosure level of other enterprises in the same industry, ambitious enterprises will disclose more comprehensive and detailed carbon information in order to gain a competitive advantage and win the support of the government, investors, and the public [35]. The pressure and threat of the external environment will affect the business behavior of peer enterprises, so the business behavior of peer enterprises will be affected by the institutional environment. Strict environmental regulation can attract the attention of peer enterprises and force them to respond actively.

From the perspective of implicit institutional pressure, Confucian culture plays a guiding role in the selection of enterprise values to a certain extent, and guides enterprises to consciously bear the responsibility of disclosing carbon emission information. Confucian culture advocates the “unity of heaven and man”. The ecological wisdom contained in Confucian culture helps enterprise managers form an awareness of green development and protecting the ecological environment. In addition, Confucian culture advocates “benevolence” and encourages people to live in harmony with nature and protect nature. A Confucian culture deeply affects the environmental protection concept of enterprise management by shaping values; therefore, enterprises that are more influenced by Confucian culture are more likely to improve their response to changes in the external environment and bear the social responsibility of disclosing carbon information. This is a concentrated reflection of enterprises actively undertaking social responsibility under the pressure of many factors such as internal industry norms and public social expectations [36]. In order to meet the requirements of consistent industry norms in the field of industry organization, enterprises need to exert their imitative ability and properly adjust the internal operation mechanism so that they conform to the accepted code of conduct in the industry and receive public support and recognition [37,38]. At this stage, considering the future development prospects of the industry, social ethics, and other requirements, some enterprises disclose carbon information voluntarily, and the government and its relevant departments will also give great support and awards, such as public commendation, government environmental protection subsidies, tax reduction and other preferential policies, which stimulates the enthusiasm of carbon information disclosure of listed enterprises. From the perspective of sustainable development theory, the ultimate goal of enterprise management is to seek long-term development. In order to ensure that reliable social resources can be obtained to earn maximum economic benefits, enterprises with long-term goals attach great importance to their own environmental performance. Through carbon information disclosure activities, they can reflect their awareness of actively undertaking social–environmental responsibility and seeking development for future generations [39].

Facing different institutional pressure, in order to obtain the support of external stakeholders and avoid possible business risks, enterprises will learn from the strategic decisions of peer enterprises and make their carbon information disclosure behavior meet the legitimacy requirements. Dimaggio and Powell (1983) describe three isomorphic processes of institution—coercive, normative, and mimetic [40]. Coercive isomorphism stems from the formal and informal pressure exerted by other organizations on which an organization depends, as well as the pressure exerted by the cultural expectations in the society in which it operates. Normative isomorphism originates from the process of specialization, including two aspects: one is the legalization and formal education at the cognitive level provided by university experts; the other is the growth and improvement of the cross-organizational talent network on which the new organizational model depends. Mimetic isomorphism stems from a recognized response to uncertainty. Mimetic isomorphism lies in the imitation and learning ability of enterprises. When the low carbon technology of Chinese enterprises is not well understood by people, when the objectives of enterprise carbon information disclosure are ambiguous and contradictory, or there is uncertainty about the impact of Confucian Culture on carbon information disclosure, enterprises may establish their own carbon information disclosure system with reference to peer enterprises. In other words, when enterprises are faced with greater institutional pressure and environmental uncertainty, enterprises will learn and imitate the carbon information disclosure behavior of peer enterprises [10,12].

In conclusion, in order to obtain policy awards from the government and avoid possible environmental penalties, peer enterprises will continuously improve their comprehensive quality and ability. Under the influence of Confucian culture, the management of peer enterprises realizes that protecting the environment is an important part of the enterprise’s business strategy. That is to say, peer enterprises have strengthened their response to changes in the external environment and industrial development and disclose higher quality carbon information. Peer enterprises pass on their concept of sustainable development to the outside world and win the trust of stakeholders [11,12,41]. Based on the above analysis, the following assumptions are proposed:

**Hypothesis** **3**.
*When explicit institutional pressure plays a role, enterprise peer influence will be significantly improved.*


**Hypothesis** **4**.
*When implicit institutional pressure plays a role, enterprise peer influence will be significantly improved.*


### 2.3. Institutional Pressure, Enterprise Peer influence, and Corporate Carbon Information Disclosure

Carbon information disclosure is a management activity of enterprises, which is the result of a comprehensive function of internal and external factors. Enterprises should not only fully consider the impact of external institutional pressure but also consider the role of competitors in the same industry when disclosing carbon information [42]. On the one hand, the government and regulatory authorities formulate relevant explicit systems and regulations to forcibly regulate the carbon information disclosure behavior of enterprises. At the same time, the implicit institutional pressure represented by Confucian culture has a subtle impact on people’s code of conduct and also indirectly affects the carbon information disclosure behavior of enterprises. On the other hand, due to the imitation and learning in enterprise carbon information disclosure, enterprises will adjust their behavior and disclosure system under the influence of peer enterprises so that the carbon information disclosure tends to the average industry level [10,40,43]. With increasingly serious environmental pollution, stakeholders expect enterprises to disclose carbon information in a comprehensive and detailed manner. When many peer enterprises disclose carbon information, keeping up with the level of carbon information disclosure in the same industry is considered an effective way to deal with external pressure. Of course, enterprises also want to imitate and learn from the leading, large-scale, and prestigious enterprises in carbon information disclosure [44,45]; however, imitation and learning have costs that are limited by social communication networks [46]. So, imitating the average disclosure of the peer enterprise has lower search costs and better peer effects. According to signaling theory, in the current capital market, the carbon information disclosure behavior of enterprises can be regarded as an external signal. For example, for those companies with better business conditions, high-level carbon information disclosure can convey their competitive advantages and good development prospects to external stakeholders through signal transmission; therefore, these enterprises are willing to disclose more carbon information [47]. For those companies with general or poor operating conditions, if they disclose some necessary carbon information reasonably, they can attract the attention of external investors by virtue of this behavior, which is conducive for enterprises to obtain the investment of enterprises in time in order to reverse the adverse profit situation, and win the opportunity to show their actual ability, and change the corporate image; therefore, regardless of the effectiveness of their business performance, each enterprise subject will pay special attention to expanding its attention to the content of carbon information disclosure. They make their carbon information disclosure behavior legal by improving their industry peer influence so as to reduce the impact of external institutional pressure [48]. According to the dynamic competition theory, enterprises improve their carbon information disclosure level by adjusting their own behavior and enhancing their learning and response level of carbon information level in the same industry in order to maintain their competitive advantage [49]. When the carbon information disclosure level of peer enterprises is high, enterprises will pay more attention to the positive contribution of carbon information disclosure to their own business performance and ecological environment in order to avoid falling behind in the competition. After keeping up with the carbon disclosure level of other enterprises in the same industry, enterprises disclose higher quality carbon information by improving the industry response level so as to reduce the impact of external explicit institutional pressure. At the same time, it is a signal to the outside world that the enterprise will have a great development prospect and strong competitive strength in the future, allowing investors to judge the fulfillment of the company’s environmental protection obligations through the company’s open carbon information so as to find that the enterprise is actually a competitive enterprise that dares to bear the environmental protection responsibilities and is very worthy of investment [50]. From this point of view, the peer influence of enterprises has a certain impact on the relationship between institutional pressure and carbon information disclosure.

For the issue of carbon information disclosure, enterprises are the main body of the implementation of the action, which can play a huge subjective initiative in the specific process of carbon information disclosure [51]. At present, due to the role of a social contract, the relationship between enterprises and external stakeholders has become closer. Under the influence of external institutional pressure, explicit institutional pressure restricts the carbon emission behavior of enterprises in the form of mandatory provisions, while implicit institutional pressure such as Confucian culture also affects the behavior choice of enterprises in people’s hearts, forming a social atmosphere of energy conservation and emission reduction. In order to show the outstanding comparative advantages of enterprises in the same industry in the fierce social competition, enterprises will take the initiative to learn from enterprises with high-level carbon information disclosure after an in-depth analysis of the current industry reality, which gradually makes their carbon information disclosure level meet the internal requirements of the industry development at this stage [52]; therefore, in order to alleviate the adverse effects of the above-mentioned pressure as soon as possible, enterprises will cater to the needs of stakeholders for information and respond to the call of the national low carbon environmental protection concept. Then, they expand the disclosure of carbon information content of enterprises and make a timely cognitive response to the institutional pressure. At the same time, enterprises will maintain the relationship between themselves and the external subjects, put an end to the phenomenon of strong exclusion of the public or fall into the disadvantaged position of severe punishment by the environmental protection department, and finally achieve the purpose of establishing a good social image for the enterprise [53,54]. In conclusion, when enterprises are faced with different institutional pressures, they will adjust and regulate their carbon information disclosure behavior clearly by learning and imitating the carbon information disclosure behavior of peer enterprises; that is, the enterprise peer influence can play a mediating role in the relationship among explicit institutional pressure, implicit institutional pressure, and corporate carbon information disclosure.

The relationship between the three can be shown in Figure 1.

Based on the above analysis, the following assumptions are proposed:

**Hypothesis** **5**.
*Enterprise peer influence has a certain mediating effect on the relationship between explicit institutional pressure and corporate carbon information disclosure.*


**Hypothesis** **6**.
*Enterprise peer influence has a certain mediating effect on the relationship between implicit institutional pressure and corporate carbon information disclosure.*


## 3. Model Construction and Data

### 3.1. Data Sources

We selected the data of A-share listed companies in China from 2014 to 2018 as the research sample. After obtaining the initial sample, the screening was started according to the following criteria: (1) Eliminate the listed companies in the financial industry; (2) eliminate ST and *ST companies; (3) eliminate the companies with a serious lack of data. Finally, the data of 920 companies were obtained, with a total of 4600 observations. The data used to measure the carbon information disclosure level of listed companies mainly come from the social responsibility report. Listed companies may issue social responsibility reports separately or in annual reports. In addition, all the social responsibility reports referred to in this paper were compiled according to the relevant guidelines issued by the Shanghai Stock Exchange and Shenzhen Stock Exchange, which can truly and objectively reflect the concrete behaviors of enterprises in fulfilling their social responsibilities. We also manually collected carbon information data from the annual reports and environmental reports published on the official websites of the Shanghai Stock Exchange and Shenzhen Stock Exchange. Explicit institutional pressure data were collected from the website of the center for public environmental research. Implicit institutional pressure data come from the CBDB and CNRDS databases. Other financial data come from the CSMAR and RESSET databases. The statistics, processing, and analysis of the data were completed by IBM SPSS Statistics 26.0 (International Business Machines Corp, Armonk, NY, USA) and Stata15.0 software (StataCorp, College Station, TX, USA).

### 3.2. Variable Descriptions

#### 3.2.1. Response Variable

Carbon information disclosure level (*CIDI_it_*): We use the content analysis method, which is the most widely used method in the research of social responsibility and information disclosure and take the carbon information disclosure index (*CIDI_it_*) as the variable to measure the level of carbon information disclosure. Specifically, when designing the evaluation indexes of carbon information disclosure levels of enterprises, we draw on the index evaluation method used by Shen et al. (2020), Li et al. (2019), and Li et al. (2017) [55,56,57]. To a certain extent, we also make a more refined division of some key indicators. The content of carbon information disclosure can be divided into three dimensions: low carbon development strategy and management, carbon emission accounting, and carbon emission governance, which are refined into 16 specific indicators [58]. Firstly, the carbon information content of the sample company is classified into three dimensions, and then the description content of the carbon information of the relevant enterprises in their respective scope is assigned with specific values. Among them, the content without disclosing and reporting carbon information can be assigned a value of 0 point; the content with a qualitative description of the specific carbon information can be assigned a value of 1 point; the content with a quantitative description of carbon information can be assigned a value of 2. Finally, we summarize the scores and the range of all scores is between 0 and 22. This paper measures CIDI by the natural logarithm of the carbon information disclosure score of listed companies. The specific evaluation items, standards, and descriptions are shown in Table 1.

#### 3.2.2. Explanatory Variables: Institutional Pressure

(1) Explicit institutional pressure (EIP): The explicit institutional pressure is essentially the performance of the government’s coercive force, which originates from various environmental protection systems and regulations issued by the state and governments at all levels. Among the existing literature studies, the common measurement methods include China’s urban pollution source supervision information disclosure index (PITI index) and the number indicators of environmental supervision laws and regulations of various industries [59,60]. The PITI index is jointly researched and developed by Institute of Public and Environmental Affairs and Natural Resources Defense Council. The objectivity of its evaluation process determines that it can be well used as an alternative variable to measure the institutional pressure on enterprise carbon information disclosure from the government and other aspects; therefore, we select the “China urban pollution source supervision information disclosure index” (PITI index) from the website of Institute of Public and Environmental Affairs to measure the explicit institutional pressure.

(2) Implicit institutional pressure (IIP): The implicit institutional pressure faced by enterprises is non-mandatory. Confucian culture is rooted in Chinese culture and has a subtle impact on the carbon information disclosure of listed companies. At present, there are some disputes about the measurement of Confucian culture in academic circles. Hilary and Hui (2009), Chen et al. (2013) measure the religious influence by the distance between the temple and the company’s registered place [61,62]. Gu (2015) measures the influence of Confucian culture by calculating the number of Confucian schools within 200 km and 300 km of the registered place of listed companies [63]. In Chinese culture, officialdom is the natural outlet for good scholars. During the Ming and Qing Dynasties, Confucian classics played an important role in the imperial examination. Jinshi refers to the successful candidate in the highest imperial examinations. In other words, Jinshi is a person with great achievements in learning Confucian culture. The number of Jinshi in a region can reflect the spread of Confucian culture. The greater the number of scholars in a region, the greater the influence of Confucian culture in the region has [64]. Referring to the research of Pan et al. (2021), we take the registered place of listed companies as the center, calculate the number of Jinshi in the Ming and Qing Dynasties within a radius of 100 km, and take the logarithm to measure the implicit institutional pressure [65]. The information of Jinshi in the Ming and Qing Dynasties comes from the CBDB and CNRDS databases, and the longitude and latitude data of the place of registration of listed companies comes from the CSMAR database.

#### 3.2.3. Mediating Variable

Enterprise peer influence level (PIL): In terms of content, there may be many areas of concern in evaluating enterprise peer influence; however, specifically, the peer influence level of enterprises in carbon information disclosure discussed here mainly refers to the positive response to the uncertainty and risk of carbon information disclosure, that is, enterprises adjust the carbon information disclosure system and mechanism by imitating and learning from peers. According to the dynamic competition theory, peer enterprises have frequent competition, interactions, and have similar development conditions. Under certain conditions, enterprises with high peer influence levels can timely identify the risks caused by environmental regulation, avoid risks in time, help enterprises deal with external shocks such as environmental regulation, and form competitive advantages in the process of strategic adjustment and resource integration. Peer enterprises play an important role in determining the quality of carbon information disclosed by companies. With the increasing government environmental regulation, firstly, enterprises with certain peer influence will pay attention to the carbon information disclosure level of other enterprises in the same industry. In order to avoid punishment and meet the legitimacy requirements, enterprises usually choose the average level of carbon information disclosure of similar competitors in the industry as the goal. Secondly, in order to gain a competitive advantage, enterprises with strong peer influence will disclose higher quality carbon information based on the carbon information disclosure level of other enterprises in the same industry. We analyze this in the robustness test; therefore, referring to the measurement method of peer influence by Aerts et al. (2006) and Grennan (2018) [10,66], we select the average value of the CIDI index of the industry (the data of selected sample enterprises should be deducted when calculating the mean value) to measure the peer influence of enterprises.

#### 3.2.4. Control Variable

On the basis of the existing research, in order to increase the rigor of the research, enterprise size, financial leverage, total asset turnover, the growth rate of operating revenue, number of directors, the proportion of independent directors, equity balance degree, Tobin q value, cumulative years of listing years, the shareholding ratio of institutional investors, and nature of ownership are added as control variables. The names, symbols, and meanings of correlated variables are shown in Table 2.

### 3.3. Model Setting

After all the variables are determined, according to the previous theoretical assumptions, in order to study the impact of institutional pressure on corporate carbon disclosure under the mediating effect of enterprise peer influence, we draw on the mediating effect test procedure of Wen et al. (2005) and constructs the following three models [67].

First of all, in order to analyze the impact of institutional pressure on corporate carbon information disclosure, we construct test model 1:(1)$$CIDI=\alpha_{0}+\alpha_{1}\times IP+\sum \alpha_{i}\times Controls_{it}+\varepsilon$$

*α*_0_ is a constant term and ε is a residual term.

Secondly, in order to analyze the impact of institutional pressure on enterprise peer influence, we construct test model 2:(2)$$PIL=\beta_{0}+\beta_{1}\times IP+\sum \beta_{i}\times Controls_{it}+\varepsilon$$

*β*_0_ is a constant term and ε is a residual term.

Finally, in order to empirically analyze whether enterprise peer influence plays a mediating effect between institutional pressure and corporate carbon information disclosure, we add enterprise peer influence as a mediating variable on the basis of model 1 and model 2, and constructs test model 3
(3)$$CIDI=\gamma_{0}+\gamma_{1}\times PIL+\gamma_{2}\times IP+\sum \gamma_{i}\times Controls_{it}+\varepsilon$$

*γ*_0_ is a constant term and ε is a residual term.

Among them, PIL represents the mediating variable enterprise peer influence; IP includes two explanatory variables, namely, explicit institutional pressure (EIP) and implicit institutional pressure (IIP); *control**s_it_* represents the control variable, including enterprise size, financial leverage, total asset turnover, the growth rate of operating revenue, number of directors, the proportion of independent directors, equity balance degree, Tobin q value, cumulative years of listing years, the shareholding ratio of institutional investors, and nature of ownership.

## 4. Analysis of Empirical Results

### 4.1. Descriptive Statistical Analysis

From the results of the descriptive statistical analysis in Table 3, it can be seen that the minimum value of CIDI is 0, and the maximum value is 2.944 (The corresponding score of carbon information disclosure is about 19), which shows that there is a big difference in the level of carbon information disclosed by listed companies in China’s A-share market, while the average value of CIDI is only 1.327 (The corresponding score of carbon information disclosure is about 3.769). It shows that the overall level of carbon information disclosure of listed companies in China’s A-share market is not high at present. Although in terms of quantity, it can be seen that more and more listed companies have begun to disclose some content related to carbon information of enterprises; however, in fact, the overall carbon disclosure of listed companies in China is still at a low level. EIP represents the explicit institutional pressure faced by enterprises, with an average value of 61.073, which indicates that China has strengthened the supervision and management of specific carbon emission activities of listed enterprises in many aspects. The minimum value of EIP is 27.800 and the maximum value is 80.800, which indicates that there are regional differences in the external explicit institutional pressure faced by listed enterprises of A-share listed companies in China. IIP indicates the implicit institutional pressure faced by enterprises, with a minimum value of 0.693 and a maximum value of 7.403, indicating that there are certain differences in the degree of influence of Confucian culture among enterprises. PIL represents the peer influence of enterprises, with an average value of 5.133, a minimum value of 1.333, and a maximum value of 11.769. It indicates that with the operating capacity of China’s listed enterprises, the internal development of China’s listed enterprises in the A-share market is generally in a positive direction, but there are still significant differences in the level of peer influence among listed enterprises.

### 4.2. Correlation Analysis

In order to test the hypothesis and further analyze the relationship between different institutional pressure and carbon information disclosure and enterprise peer influence, the Pearson correlation test is also conducted, and the test results are shown in Table 4.

It can be seen from Table 4 that the correlation coefficients between the response variable CIDI, the explanatory variable EIP, IIP, and the mediating variable PIL are 0.029, 0.047, and 0.372, respectively, which are significant at the level of 5%. This shows that there is an obvious positive correlation between the level of enterprise carbon information disclosure and explicit institutional pressure, implicit institutional pressure, and enterprise peer influence; that is, when enterprises are affected by different external institutional pressures, their carbon disclosure level has increased accordingly to some extent. The above results are basically consistent with hypotheses 1 and 2. According to the test results of the variance inflation factor in Table 5, VIF values are less than 10, indicating that there is no multicollinearity problem between variables.

### 4.3. Multiple Regression Analysis

Based on the above discussion and analysis, in order to further test the proposed hypothesis and verify the correlation among variables, we conduct multiple linear regression analyses on the samples. The regression results are shown in Table 6.

Model 1 tests the impact of institutional pressure on enterprise carbon information disclosure. From Table 6, it can be seen that in the study of explicit institutional pressure, the adjusted R^2^ of model 1 is 0.345, and the F value is 78.74 at a 1% significance level. The beta coefficient of EIP is 0.003, and *p* value is significant at the level of 1%, which indicates that the level of carbon information disclosure of A-share listed companies has been improved after being affected by the mandatory pressure of government laws, regulations, and policies. That is to say, the explicit institutional pressure will have a positive impact on the improvement of the carbon information disclosure level of enterprises; thus, hypothesis 1 is verified. Similarly, in the study of implicit institutional pressure, the adjusted R^2^ of model 1 is 0.355, and the F value is 74.55 at a 1% significance level. The beta coefficient of IIP is 0.023, and *p* value is significant at the level of 1%. This shows that the greater the impact of Confucian culture on enterprises, the higher the level of carbon information disclosure; that is, when enterprises bear a certain range of implicit institutional pressure, they will play a positive role in promoting the level of carbon information disclosure. Hypothesis 2 is verified. The result is consistent with those of Tang et al. (2020) and Tang et al. (2021) [58,68].

Model 2 tests the influence of institutional pressure on the peer influence of enterprises. From Table 6, on the one hand, for the explicit institutional pressure, the adjusted R^2^ of model 2 is 0.854, and the F value is 819.70 at a 1% significance level, indicating that the fitting degree of the model is appropriate. The beta coefficient of EIP is 0.016, and *p* value is significant at the level of 1%. This indicates that the self-awareness of listed companies in the same industry will be affected by external explicit institutional pressure, and the greater the pressure is, the higher the level of enterprise peer influence will be. In other words, the influence of explicit institutional pressure on the peer influence of enterprises is significantly positive; thus, hypothesis 3 is verified. On the other hand, for implicit institutional pressure, the adjusted R^2^ of model 2 is 0.895, and the F value is 1631.56 at a 1% significance level, indicating that the fitting degree of the model is good. The beta coefficient of IIP is 0.020, and *p* value is significant at the level of 1%. This shows that the peer influence of listed enterprises in China will also change positively due to the influence of Confucian culture; that is, when the implicit institutional pressure faced by enterprises plays a role, the industry peer influence of enterprises will be significantly improved, and Hypothesis 4 has been verified; therefore, both external institutional pressures have a very significant positive impact on the peer influence of enterprises.

In order to further test hypothesis 5 and hypothesis 6, model 3 takes enterprise peer influence as a mediating variable to further test the mediating effect of enterprise peer influence between different institutional pressures and enterprise carbon information disclosure. According to the mediating effect test steps of Wen et al. (2005)’s research, as shown in Figure 2 [67], we first test coefficient α_1_, that is, the regression coefficient corresponding to the institutional pressure in model 1. According to the empirical results of model 1, it is known that both the explicit institutional pressure and the implicit institutional pressure have a significant positive effect on the enterprise carbon information disclosure level; therefore, the regression coefficient α_1_ of the institutional pressure and carbon information disclosure is significant. Then, we test the coefficients β_1_ and γ_1_, in which β_1_ refers to the regression coefficient to which the institutional pressure corresponds in model 2. γ_1_ refers to the regression coefficient of the mediating variable enterprise peer influence in the mediating model of enterprise peer influence to institutional pressure and corporate carbon information disclosure. According to the empirical results of model 2, there is a significant positive correlation between the explicit institutional pressure, the implicit institutional pressure, and the peer influence of enterprises; that is, the regression coefficient β_1_ of institutional pressure on the peer influence of enterprises is significant. At the same time, in model 3, under the effect of explicit institutional pressure and implicit institutional pressure, the beta coefficients of PIL are 0.107 and 0.102, respectively, and *p* value is significant at the level of 1%, which means that the coefficient γ_1_ of mediator variable PIL is significant in the mediating effect test model of institutional pressure and corporate carbon information disclosure. Under the condition that the empirical results of the coefficients β_1_ and γ_1_ are significant, the last procedure of the mediating effect test is performed to test the coefficient γ_2_; that is, to test the coefficient corresponding to the institutional pressure in the mediating effect test model of enterprise peer influence on institutional pressure and carbon information disclosure. From the results of model 3, it can be seen that the beta coefficients of EIP and IIP are 0.001 and 0.018. The *p* value of explicit institutional pressure is not significant, and the *p* value of implicit institutional pressure is significant at the level of 5%; therefore, the enterprise peer influence level has a certain mediating effect on the relationship between external institutional pressure and corporate carbon information disclosure. According to the results of model 3, after adding the mediating variable peer, the coefficients of EIP and IIP decreased compared with model 1. Specifically, enterprise peer influence plays a full mediating effect in the relationship between explicit institutional pressure and enterprise carbon information disclosure. Enterprise peer influence plays a partial mediating effect in the relationship between implicit institutional pressure and corporate carbon information disclosure. This means that when facing external institutional pressure, peer enterprises make the decision to improve carbon information disclosure. Carbon information disclosure has peer influence. The improvement of enterprise peer influence level promotes the improvement of the overall carbon information disclosure level of the industry.

### 4.4. Robustness Test

In order to make the research results stable, we also carry out a robustness test. The specific contents are as follows: First, the explanatory variable explicit institutional pressure is replaced by the number of environmental protection administrative punishment cases in the province where the enterprise is registered in that year. Second, the explanatory variable implicit institutional pressure is replaced by the number of Confucian academies in the city where the enterprise is registered. Confucianism is mainly spread in the form of education in academies; therefore, the number of Confucian academies can reflect the influence of Confucian culture in this area to a certain extent. Third, considering peer influence in enterprise carbon information disclosure, there is a situation of learning and imitating from the disclosure leading enterprises; this paper changes the measurement method of enterprise peer influence. In the context of increasing environmental regulation, in order to obtain legitimacy, recognition, and reputation, enterprises will consciously keep up with the level of carbon information disclosure in the same industry. In order to gain a competitive advantage, enterprises with strong peer influence will disclose more carbon information than enterprises in the same industry. Thus, we adopt “the enterprise’s current carbon information disclosure level—the average of the industry’s carbon information disclosure level in the previous period” to measure enterprise peer influence. In addition, the regression analysis is carried out by following the test steps in turn, and the final robustness results are shown in Table 7.

From the robustness results in Table 7, in the study of explicit institutional pressure (EIP), the values of coefficients α_1_, β_1_, γ_1,_ and γ_2_ are 0.063, 0.192, 0.175, and 0.011, respectively, and the coefficients α_1_, β_1,_ and γ_1_ are significant at the level of 1%. Similarly, in the study of implicit institutional pressure (IIP), the values of coefficients α_1_, β_1_, γ_1,_ and γ_2_ are 0.071, 0.135, 0.179, and 0.027, respectively, and the coefficients α_1_, β_1_, γ_1,_ and γ_2_ are significant at the level of 5%. It shows that after changing the measurement method of main variables, peer influence level can still produce a significant mediating effect between different institutional pressure and enterprise carbon information disclosure, which is consistent with the above research results. It proves that the empirical research conclusions meet the requirements of robustness.

## 5. Conclusions

We selected 920 listed companies in China’s A-share market from 2014 to 2018 as research samples and studied the influence of institutional pressure on corporate carbon information disclosure, the influence of institutional pressure on enterprise peer influence, and the mediating effect of enterprise peer influence between institutional pressure and corporate carbon information disclosure. Combined with the specific theoretical basis analysis and empirical results test, the six hypotheses proposed above are verified, respectively, and the following research conclusions are concluded:

(1) Institutional pressure, namely environmental regulation and Confucian culture, can significantly improve the level of carbon information disclosure of listed enterprises. On the one hand, in the daily operation of enterprises, the more pressure they feel from an explicit institution such as laws, regulations, and policies, the more stringent the regulatory environment they face, and the more attention they pay to their disclosure of carbon information. On the other hand, Confucian culture has a subtle impact on enterprises. It indirectly affects the carbon information disclosure behavior of enterprises by influencing people’s code of conduct and business philosophy of enterprises. Listed companies affected by Confucian culture tend to actively assume social responsibility and actively respond to environmental protection requirements; therefore, they will reasonably disclose the relevant carbon information content of the company to better meet the needs of social stakeholders.

(2) Institutional pressure can obviously stimulate enterprises to enhance their peer influence. Under normal circumstances, any enterprise will inevitably suffer from the direct effect of the explicit external institution and the potential constraint of the implicit internal institution of the industry. In the face of this dual pressure, in order to survive for a long time, enterprises must strictly require themselves to continuously improve their comprehensive quality and give full play to their peer influence in the industry. In addition, enterprises have to take the initiative to adapt to the external institutional environment and take appropriate measures to adjust the current behavior in order to maintain their core competitiveness.

(3) There is a certain significant mediating effect of enterprise peer influence between two different institutional pressure and corporate carbon information disclosure. Under the influence of explicit external institutional pressure, enterprises will imitate and learn from the carbon information disclosure behavior of peer enterprises, therefore the mediating effect of enterprise peer influence is very obvious. Based on the motivation of seeking advantages and avoiding disadvantages, the improvement of the carbon information disclosure level of peer enterprises can promote the level of carbon information disclosure of enterprises as a whole. On the one hand, it shows that the explicit institutional pressure that scholars have studied before does have a mandatory restriction on corporate carbon information disclosure. On the other hand, it also tells us that we must not ignore the auxiliary binding force which the implicit institutional pressure brings to corporate carbon information disclosure.

## 6. Implications

### 6.1. Develop and Improve Carbon Disclosure Institution

From the aspect of explicit institutional pressure, there is a serious lack of laws and regulations on information disclosure in China, and a unified carbon accounting institution has not been formed; therefore, we suggest that the government and regulatory authorities shall refer to the international regulations on environmental accounting information disclosure and formulate clear and specific requirements on the way and content of carbon information disclosure. For example, in the way of carbon information disclosure, the social responsibility report provided by enterprises should include not only the carbon information content described quantitatively in monetary form but also the qualitative description and analysis in non-monetary form so as to realize the transformation from simply using qualitative carbon disclosure to combining qualitative and quantitative carbon disclosure as soon as possible.

### 6.2. Strengthen External Supervision through the Joint Participation of All Sectors of Society

We need to rely on the potential implicit institution to guide corporate behavior in a moderate and orderly way by virtue of unwritten implicit means such as ethics and values. For example, the government can provide financial subsidies, honorary title awards, or launch some benign incentive forms to enterprises with good quality carbon information disclosure. In addition, through the participation of all social subjects, we can effectively supervise the carbon information disclosure of enterprises, so that listed enterprises can pay attention to the impact of external supervision mechanisms, consciously fulfill their environmental protection responsibilities, and regularly publish carbon information disclosure reports.

### 6.3. Encourage Enterprises to Actively Undertake Environmental Protection Responsibilities

Enterprises are the main users of social resources in China. According to the principle of “who pollutes, who governs”, enterprises certainly have to shoulder the responsibility of environmental protection. In the production and operation activities, we should first have an awareness of carbon information disclosure and formulate a detailed mechanism to measure the carbon emission of enterprises. Secondly, enterprises should strengthen the construction of their own internal development peer influence. In addition to improving the internal comprehensive management ability of enterprises, we should actively improve the quality of carbon information disclosure, accelerate the green transformation, and gradually keep pace with the leaders in the industry. Finally, the carbon information disclosure level of listed companies in China will reach a new level as a whole.

## Figures and Tables

**Figure 1 ijerph-19-04174-f001:**
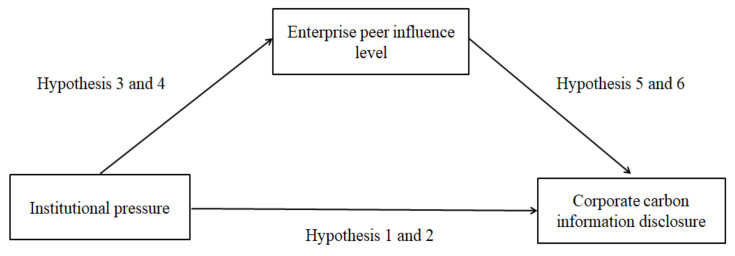
Relationship among institutional pressure, enterprise peer influence, and corporate carbon information disclosure.

**Figure 2 ijerph-19-04174-f002:**
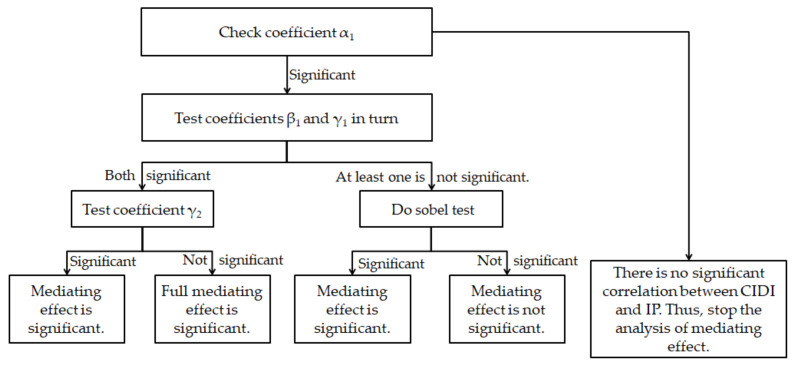
Test procedure of mediating effect. Note: *α*_1_ represents the regression coefficient corresponding to the institutional pressure in model 1 of institutional pressure and corporate carbon information disclosure; *β*_1_ refers to the regression coefficient corresponding to institutional pressure in model 2 of institutional pressure and enterprise peer influence; *γ*_1_ refers to the regression coefficient of the mediating variable enterprise peer influence in of the mediating effect test of enterprise peer influence on institutional pressure and corporate carbon information disclosure (model 3); *γ*_2_ refers to the regression coefficient of explanatory variable institutional pressure in the mediating effect test of enterprise peer influence on institutional pressure and corporate carbon information disclosure (model 3).

**Table 1 ijerph-19-04174-t001:** Evaluation index of carbon information disclosure level.

Dimensions of Carbon Information Disclosure	Carbon Information Disclosure Indicators	Explanation
Low carbon development strategy and management	Whether to integrate carbon emission reduction into enterprise development strategy	1 point for incorporating carbon emission reduction into enterprise development strategy; otherwise 0.
	Whether to set carbon emission reduction targets	1 point for setting clear carbon emission reduction targets; otherwise 0.
	Risks of carbon reduction	1 point for disclosing the business risks and economic benefit losses caused by the government’s carbon emission regulation; otherwise 0.
	Establish low carbon management organization or system	1 point for setting up a special low carbon management organization or establishing a low carbon management system; otherwise 0.
	Enhance employees’ low carbon awareness	1 point for carrying out low carbon environmental protection publicity for enterprise employees; otherwise 0.
	Integrate carbon emission reduction into reward, punishment, and assessment mechanism	In the rules and regulations of the enterprise, if the reward and punishment management mechanism for carbon emission reduction is clearly formulated, it can be assigned as 1 point; otherwise 0.
Carbon emission accounting	Annual carbon emissions of enterprises	1 point for qualitative description and 2 points for quantitative description; otherwise 0.
	Carbon emission classification and accounting standards	1 point for qualitative description and 2 points for quantitative description; otherwise 0.
	Energy consumption of the enterprise in the current year	1 point for qualitative description and 2 points for quantitative description; otherwise 0.
	Whether the enterprise has passed ISO14001 environmental management system certification	1 point for passing, otherwise 0.
Carbon emission governance	Carbon emission reduction of enterprises	1 point for qualitative description and 2 points for quantitative description; otherwise 0.
	Whether enterprises invest in technology, capital, and labor for carbon reduction	The enterprise has carried out technology research and development and project investment for energy conservation and emission reduction, such as the new purchase, reconstruction, and expansion of environmental protection equipment or projects, and low carbon technology innovation activities. 1 point for qualitative description and 2 points for quantitative description; otherwise 0.
	Cost of daily carbon emission reduction	1 point for the disclosure of sewage charges, greening fees, and other expenses incurred in the process of carbon emission reduction management; otherwise 0.
	Benefits from carbon emission reduction.	1 point for qualitative description and 2 points for quantitative description; otherwise 0.
	Government support for carbon reduction	1 point for obtaining government financial subsidy support; otherwise 0.
	Administrative penalties for environmental protection	1 point for not being punished by environmental protection administration; otherwise 0.

**Table 2 ijerph-19-04174-t002:** Definition of variables.

Variable Type	Variable Name	Variable Symbol	Definition and Description
Response Variable	Carbon information disclosure index	CIDI	Natural logarithm of carbon information disclosure score
Explanatory variable	Explicit institutional pressure	EIP	Pollution Information Transparency Index (PITI)
	Implicit institutional pressure	IIP	Confucian culture, that is, the logarithm of the number of Jinshi in the Ming and Qing dynasties within 100 km of the registered area of listed companies
Mediating variable	Peer influence level of enterprises	PIL	The average value of CIDI after the sample enterprises are eliminated
Control variable	Enterprise size	Size	Natural logarithm of total assets at the end of the period
	Financial leverage	LEV	Total liabilities at the end of the year/total assets at the end of the year
	Total asset turnover	ATO	Operating income/average total assets
	Growth rate of operating revenue	Growth	Current year’s operating income/previous year’s operating income-1
	Number of directors	Board	Natural logarithm of board members
	Proportion of independent directors	INDEP	Number of Independent Directors/the number of directors
	Equity balance degree	Balance	The sum of the shareholding ratio of the second to fifth largest shareholders/the shareholding ratio of the first largest shareholder
	Tobin Q value	Tobin Q	(Current stock market value + number of non-tradable shares) × Net assets per share + book value of liabilities)/total assets
	Cumulative years of listing years	List	Natural logarithm of cumulative listing years
	Shareholding ratio of institutional investors	INST	Total shares held by institutional investors/circulating share capital
	Nature of ownership	OWN	The value of state-owned holding enterprise is 1; otherwise 0

**Table 3 ijerph-19-04174-t003:** Descriptive statistical analysis results.

Variable	Sample Size	Minimum	Maximum	Mean	Standard Deviation
CIDI	4600	0	2.944	1.327	0.964
EIP	4600	27.800	80.800	61.073	12.328
IIP	4600	0.693	7.403	5.484	1.597
PIL	4600	1.333	11.769	5.133	2.110
Size	4600	20.233	26.063	22.465	1.274
LEV	4600	0.068	0.859	0.423	0.192
ATO	4600	0.109	2.129	0.638	0.358
Growth	4600	−0.399	1.337	0.144	0.276
Board	4600	1.609	2.708	2.135	0.205
INDEP	4600	0.333	0.571	0.376	0.053
Balance	4600	0.037	2.800	0.681	0.585
Tobin Q	4600	0	7.788	2.076	1.296
List	4600	1.099	3.258	2.361	0.551
INST	4600	0.005	0.870	0.422	0.233
OWN	4600	0	1	0.393	0.488

**Table 4 ijerph-19-04174-t004:** Correlation analysis results.

Variables	CIDI	EIP	IIP	PIL	Size	LEV	ATO	Growth	Board	INDEP	Balance	Tobin Q	List	INST	OWN
CIDI	1														
EIP	0.029 **	1													
IIP	0.047 ***	0.358 ***	1												
PIL	0.372 ***	−0.004	0.002	1											
Size	0.466 ***	0.083 ***	0.020	0.271 ***	1										
LEV	0.187 ***	−0.040 ***	−0.057 ***	0.100 ***	0.537 ***	1									
ATO	0.104 ***	0.060 ***	0.041 ***	0.100 ***	0.074 ***	0.118 ***	1								
Growth	−0.049 ***	0.076 ***	−0.014	−0.002	0.024 *	−0.001	0.133 ***	1							
Board	0.223 ***	−0.072 ***	0.041 ***	0.128 ***	0.291 ***	0.187 ***	−0.003	−0.018	1						
INDEP	−0.067 ***	0.004	−0.066 ***	−0.024	0.017	0.012	−0.038 ***	−0.033 **	−0.533 ***	1					
Balance	−0.056 ***	0.051 ***	−0.006	−0.063 ***	−0.102 ***	−0.112 ***	−0.117 ***	0.071 ***	0.018	0.008	1				
Tobin Q	−0.215 ***	−0.013	−0.012	−0.179 ***	−0.464 ***	−0.370 ***	−0.005	0.003	−0.149 ***	0.010	0.007	1			
List	0.288 ***	−0.062 ***	−0.066 ***	0.249 ***	0.345 ***	0.333 ***	0.119 ***	−0.113 ***	0.199 ***	−0.053 ***	−0.158 ***	−0.160 ***	1		
INST	0.293 ***	−0.008	0.023	0.170 ***	0.447 ***	0.201 ***	0.146 ***	−0.049 ***	0.230 ***	−0.070 ***	−0.219 ***	−0.013	0.287 ***	1	
OWN	0.273 ***	−0.152 ***	0.005	0.173 ***	0.374 ***	0.321 ***	0.055 ***	−0.123 ***	0.294 ***	−0.050 ***	−0.253 ***	−0.171 ***	0.530 ***	0.418 ***	1

Note: * *p* < 0.1, ** *p* < 0.05, *** *p* < 0.01.

**Table 5 ijerph-19-04174-t005:** Test of variance inflation factor.

Variables	Size	OWN	Board	LEV	List	INST	INDEP	Tobin Q	EIP	IIP	PIL	Balance	ATO	Growth	Mean
VIF	2.25	1.77	1.72	1.56	1.54	1.54	1.50	1.41	1.23	1.18	1.14	1.12	1.09	1.06	1.44
1/VIF	0.444	0.566	0.580	0.641	0.648	0.649	0.668	0.710	0.813	0.850	0.878	0.892	0.919	0.943	

**Table 6 ijerph-19-04174-t006:** Results of multiple linear regression analysis.

Variable	Explicit Institutional Pressure			Implicit Institutional Pressure		
	CIDI	PIL	CIDI	CIDI	PIL	CIDI
	Model 1	Model 2	Model 3	Model 1	Model 2	Model 3
EIP	0.003 ***	0.016 ***	0.001			
	(2.93)	(16.01)	(0.75)			
IIP				0.023 ***	0.020 ***	0.018 **
				(3.16)	(3.13)	(2.49)
PIL			0.107 ***			0.102 ***
			(18.27)			(16.77)
Size	0.295 ***	−0.059 ***	0.283 ***	0.301 ***	0.028 **	0.287 ***
	(21.59)	(−3.81)	(20.62)	(22.19)	(2.18)	(21.03)
LEV	−0.389 ***	−0.277 ***	−0.507 ***	−0.322 ***	0.279 ***	−0.481 ***
	(−4.74)	(−3.36)	(−6.49)	(−3.94)	(3.94)	(−6.13)
ATO	0.041	−0.158 ***	0.147 ***	0.049	−0.065 *	0.144 ***
	(1.01)	(−3.70)	(4.12)	(1.19)	(−1.81)	(4.00)
Growth	−0.164 ***	0.273 ***	−0.178 ***	−0.188 ***	0.099 **	−0.182 ***
	(−3.80)	(6.19)	(−4.08)	(−4.31)	(2.50)	(−4.10)
Board	0.191 ***	−0.147 *	0.197 ***	0.200 ***	−0.280 ***	0.202 ***
	(2.75)	(−1.84)	(2.84)	(2.85)	(−4.05)	(2.88)
INDEP	−0.516 **	0.437	−0.623 **	−0.559 **	−0.043	−0.613 **
	(−2.01)	(1.56)	(−2.39)	(−2.18)	(−0.19)	(−2.35)
Balance	0.045 **	0.033	0.049 **	0.038 *	−0.022	0.047 **
	(2.20)	(1.61)	(2.35)	(1.87)	(−1.33)	(2.25)
Tobin Q	−0.028 ***	−0.145 ***	−0.007	0.000	−0.003	0.005
	(−2.60)	(−13.24)	(−0.64)	(0.01)	(−0.37)	(0.46)
List	0.172 ***	0.526 ***	0.133 ***	0.107 ***	−0.081 ***	0.121 ***
	(6.51)	(19.57)	(5.19)	(3.81)	(−3.18)	(4.48)
INST	0.258 ***	0.018	0.198 ***	0.241 ***	−0.064	0.193 ***
	(3.94)	(0.28)	(3.06)	(3.71)	(−1.23)	(2.99)
OWN	0.128 ***	−0.229 ***	0.097 ***	0.151 ***	0.068 **	0.102 ***
	(3.91)	(−6.36)	(3.03)	(4.65)	(2.30)	(3.22)
Constant	−6.571 ***	1.218 ***	−6.123 ***	−6.644 ***	3.150 ***	−6.267 ***
	(−17.58)	(3.10)	(−18.78)	(−17.74)	(9.28)	(−19.23)
Adjusted R-squared	0.345	0.854	0.310	0.355	0.895	0.312
F-statistic	78.74 ***	819.70 ***	217.68 ***	74.55 ***	1631.56 ***	170.98 ***

Note: *t*-statistics in parentheses; * *p* < 0.1, ** *p* < 0.05, *** *p* < 0.01.

**Table 7 ijerph-19-04174-t007:** Results of robustness analysis.

Variable	Explicit Institutional Pressure			Implicit Institutional Pressure		
	CIDI	PIL	CIDI	CIDI	PIL	CIDI
	Model 1	Model 2	Model 3	Model 1	Model 2	Model 3
EIP	0.063 ***	0.192 ***	0.011			
	(4.78)	(2.67)	(1.05)			
IIP				0.071 ***	0.135 **	0.027 ***
				(5.84)	(2.24)	(3.43)
PIL			0.175 ***			0.179 ***
			(70.68)			(64.76)
Size	0.295 ***	1.178 ***	0.103 ***	0.303 ***	1.332 ***	0.111 ***
	(21.88)	(15.54)	(9.74)	(18.45)	(14.29)	(8.73)
LEV	−0.385 ***	−2.024 ***	−0.340 ***	−0.321 ***	−1.305 ***	−0.239 ***
	(−4.68)	(−5.19)	(−5.94)	(−3.44)	(−2.79)	(−3.64)
ATO	0.036	0.767 ***	0.077 **	0.032	0.265	0.088 ***
	(0.88)	(3.79)	(2.55)	(0.67)	(1.01)	(2.61)
Growth	−0.174 ***	−0.904 ***	−0.048	−0.130 ***	−1.059 ***	−0.055
	(−4.04)	(−4.05)	(−1.41)	(−2.67)	(−4.24)	(−1.44)
Board	0.186 ***	0.828 **	0.125 **	0.199 **	1.315 ***	0.150 **
	(2.68)	(2.11)	(2.20)	(2.37)	(2.75)	(2.26)
INDEP	−0.494 *	−1.501	−0.303	−0.331	−0.890	−0.038
	(−1.93)	(−1.09)	(−1.59)	(−1.08)	(−0.54)	(−0.17)
Balance	0.046 **	0.322 ***	−0.015	0.050 **	0.270 **	−0.001
	(2.24)	(3.00)	(−0.99)	(2.09)	(2.21)	(−0.03)
Tobin Q	−0.025 **	0.047	−0.038 ***	−0.022 *	0.058	−0.010
	(−2.34)	(1.02)	(−4.94)	(−1.71)	(0.94)	(−1.07)
List	0.178 ***	0.294 **	0.163 ***	0.168 ***	0.437 **	0.081 ***
	(6.73)	(1.98)	(7.90)	(5.43)	(2.39)	(3.31)
INST	0.258 ***	0.665 **	0.133 ***	0.167 **	0.972 ***	0.084 *
	(3.94)	(2.09)	(2.92)	(2.25)	(2.68)	(1.65)
OWN	0.142 ***	0.519 ***	0.016	0.104 ***	0.585 ***	0.072 ***
	(4.33)	(2.97)	(0.66)	(2.81)	(2.88)	(2.73)
Constant	−6.934 ***	−29.934 ***	−1.565 ***	−6.891 ***	−34.332 ***	−1.659 ***
	(−18.17)	(−15.08)	(−5.70)	(−14.87)	(−15.08)	(−5.53)
Adjusted R-squared	0.347	0.167	0.706	0.328	0.185	0.717
F-statistic	79.73 ***	57.29 ***	705.5 ***	66.15 ***	14.74 ***	469.5 ***

Note: *t*-statistics in parentheses; * *p* < 0.1, ** *p* < 0.05, *** *p* < 0.01.

## Data Availability

Not applicable.

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
