# Peer review of "A Study on the Impact of Institutional Pressure on Carbon Information Disclosure: The Mediating Effect of Enterprise Peer Influence"

_ijerph, 2022, doi:10.3390/ijerph19074174_

Round 1
Reviewer 1 Report
Thank you for the opportunity to review the manuscript The Influence of Institutional Pressure on Carbon Information Disclosure—A Research on the Mediating Effect of Enterprise Cognitive Ability. The paper offers an interesting and timely perspective on corporate carbon disclosure practices, informed by the unique perspective of Chinese cultural influences on social and economic development. The introduction of enterprise (corporate) cognitive ability is a provocative variable and one that the authors make interesting use of throughout their manuscript.
That being said, the manuscript is not without challenges however. This review necessitates that I locate my own position as a westerner, largely unfamiliar with the context of Chinese corporate culture, so these comments come from this perspective, for better or for worse. However, to some extent, assuming this paper is intended for an international audience who, we may assume, may also not be familiar with Chinese corporate culture either, then to some extent, these may be valid issues the authors should address for a non-Chinese readership.
Overview:
(1) It is unclear the extent to which the authors take corporate 'greenwashing' and deception into account when referring to the disclosure reports. This is exacerbated by the authors' use of legitimacy theory where corporate actors have every incentive to gloss over externalised and less proximal supply chain carbon loads which are excluded from their own reporting parameters?
Put differently, the authors appear to have taken the reporting behaviour of corporations at face value, despite significant evidence to the contrary, especially for those MNCs which co-exist outside of the cultural influences of Confucianism, for example. Stating, as they do, in Lines 171-173, that "the Confucian thought of good faith helps to curb the false statements of enterprises and promote enterprises to disclose the real carbon emissions of the company in a timely and objective manner" is noble, but also naive given the profits and reputational benefits for being 'green' that can be accrued by externalising carbon emissions and other costs. This also leads to a tautological argument: enterprises are thought to accurately report carbon disclosures because of the implicit pressures of cultural imperatives to do so, consequently because there are such cultural imperatives, enterprises are assumed to comply.
It is reasonable, of course, that for Chinese enterprises the situation is quite different, but it is also reasonable that, when writing for an international audience, this be made more explicit about how differences between Chinese enterprises and non-Chinese enterprises manifest, such that the former can be instinctively trusted and taken at face value, while the latter can/ should not on the basis of the available evidence of 'greenwashing', etc.
In sum then, the authors are asked to clarify how these assumptions can be validated and what methods are used for doing so. Further, how do the Confucian principles work, pragmatically, to positively influence the honest reporting behaviour of enterprises.
(2) The authors do not specify what carbon information disclosure protocols are in place, despite many references to the practice which seems to be presented, and uncritically discussed, as a bit of a black box. As these protocols vary, and some measures are contested in terms of time scale, responsibility/ ownership and accounting processes, the authors are asked to clarify what mechanisms of accounting they are referring to when they refer to carbon information disclosure. Do these include carbon off-setting schemes, for example, and if so, how are such schemes themselves assessed and validated for compliance and legitimacy?
(3) The measurement of cultural imperatives and influences is rightly flagged as difficult to achieve, and the authors have proposed that the relative influence of Confucianism in effect in a region is derived from the number of Jinshi scholars in that region, and this is used as an indicator for the degree of 'spread' of Confucianism. However, for non-Chinese readers this is opaque. Specifically, how does the number of advanced scholars in a region equate to an indicator of the spread of cultural traditions and mores? This is not clear. Since this is an important part of their calculations, this needs to be better elaborated, since this appears to be resting, and is developed on the basis of some assumptions which cannot, in themselves, be tested for validity.
(4) Between Lines 387 and 393, two ideas are proposed as an account for enterprise cognitive ability, viz. Lines 387-389: "the cognitive ability of enterprises discussed here mainly refers to the timely response of an enterprise subject to the carbon information disclosure behavior of other enterprises in the same industry", and Lines 391-393: "the consideration of the cost-benefit principle, usually, enterprises will choose the average level of carbon information disclosure of other competitors in the industry as the benchmark to adjust their behavior in time".
The first segment cited refers to timely responses, although how this is measured in practice is not stated. What constitutes a 'timely response' in this context? How can this be specified and measured? What evidence is used to characterise a response and how is it determined to be timely? The second segment also refers to 'time', but again this is not defined - what reporting periods are being considered here? This also becomes more problematic insofar as carbon emissions are subject to different scales of time, including lags between implementing changes in business practice and when those changes in emissions manifest and are registered. How does timeliness as it is used here as a component of the 'cognitive ability' variable account for legacy emissions, those historical discharges that are still 'in the pipeline'?
Clarity:
Phrasing lines 33-35: Currently reads "More and more people, institutional investors and media from all walks of life have begun to pay attention to the environmental cost behind the development of enterprises". This would read better if it were rephrased, e.g., " More and more institutional investors, media, and people from all walks of life have begun to pay attention to the environmental cost behind the development of enterprises"
Lines 42-43: "External institutional pressure can be divided into explicit institution and implicit institution" - reads as an incomplete sentence, perhaps add a missing 'pressure' at the end of the sentence?
Lines 56-57: "traditional culture can virtually regulate the carbon information disclosure behavior of enterprises" within the context of Confucian culture. The authors' brief synopsis of Confucian culture is clear and well articulated. However, for a western reader it would help to elaborate a little (a sentence or two) on how such a cultural tradition "virtually" regulates the carbon information disclosure. There is a jump from an exposition of the Confucian culture to the virtual regulation of the culture which is not clear. For example, is the following paragraph an elaboration on this process of virtual regulation or not?
Line 61: What are "leading subjects in the industry"? This term has yet to be clarified, especially when in Line 63 we now find "different subjects from all walks of life". These terms warrant clarification to aid reader comprehension.
Lines 70-71: "explicit institutions, and pay less attention to the impact of potential informal institutions" (These should be plural, passim)
Lines 93-94: "the social responsibility report of China's A-share listed companies as the research sample." Can the authors state why they selected this as a sample of what population? Because these represent major corporate enterprises in China, is my suspicion, but please state this explicitly if this is (or isn't) the case, especially since doing so is given as the first of three research innovations. Moreover, in Line 102, the authors write "the research object is different from that in the past", and this requires elaboration. What makes this object different?
Lines 118-119: "the requirements of the public on the carbon information disclosure of enterprises are also increasing" - the meaning of this is not clear.
Author Response
Dear reviewer
Thank you for your valuable comments. We have revised the manuscript item by item. Please see the attachment to get the point-by-point response.
Best regards
Yours sincerely
Yongjun, Tang

Reviewer 2 Report
This proposal presents interesting research about how the institutional pressure has an influence on firm information disclosure about carbon emissions.
Authors develop a theoretical framework to ground their hypothesis. It is interesting the distinction between internal and external institutional pressure. Other papers have differentiated those approaches with another terminology, like mandatory and not mandatory public policies, which could have some parallelism with internal and external institutional pressures. Maybe, conclusions of this proposal could be enriched by explaining results comparing with previous research in this sense. There are other studies with samples in other countries that analyze differences on firm environmental adaptation depending on the policy model adopted and self-regultation, which maybe could be compared in some sense to internal pressure, has the greatest environmental benefits.
My main concerns about the paper are about the measurement of the Enterprise Cognitive Ability variable and about the correlation analysis.
In his conception, Enterprise Cognitive Ability is a firm variable, but it is measured by the average value of the CIDI index of the industry. This type of measurement implies that the arguments of the hypothesis must change, or that the variable must be measured in another way. Or maybe the variable represents another concept and not the firm cognitive ability. At least, authors should better justify this measurement.
Regarding the correlation analysis, the extremely high correlation between EIP and CIDI seems to invalidate the analysis, it shows that both variables are almost the same… so this can create problems in the regression analysis. It is necessary to corroborate assumptions that the subsequent model analysis requires. There are also another two correlations that are also high, above 0.6, so additional tests or a change in the measurement is needed.
Author Response
Dear reviewer
Thank you for your valuable comments. We have revised the manuscript item by item. Please see the attachment to get the point-by-point response. Through revision, we learn a lot from you the experience of how to improve the quality of a manuscript.
Thank you again for your insightful comments.
Best regards
Yours sincerely
Yongjun, Tang
Jun Zhu
Wenchao Ma
Mengxue Zhao
